# "AI is not Just a Technology"
# Towards an Integrated Teaching Approach to Machine Learning

**Nicole Coleman**[* 1]  **Claudia Engel**[1]

## Abstract

Reporting on our experiences introducing a broad range of staff of academic libraries to AI, we suggest that training in practical applications of AI requires more than learning the technology. AI projects in libraries require contributions by computing specialists, subject specialists, metadata librarians, conservators, and others; each bringing different expertise and competencies to the project. We offer an integrated approach to knowledge building in AI that considers the processes and roles involved. This approach takes a comprehensive view of the entire AI project from understanding the need, translating that need into tasks, identifying and preparing the data, applying methods, and evaluating the results.

## 1. Introduction

Research in AI and related subfields (such as computer vision and natural language processing) has skyrocketed over the last decade and become an increasingly important focus in computer science. In this context, academic libraries play an interesting role for two reasons. First, there is significant overlap between the traditional information management that goes on in libraries and the essential functions of classification and pattern-matching in AI. Also, libraries have invested significant resources in recent decades into digitizing efforts and become hubs for providing large amounts of digital, i.e. computationally accessible data. In addition to books, digitized library holdings range from manuscripts, to audio recordings, to notebooks from field studies, and extensive image collections. Large institutions have undertaken huge digitization projects (Wellcome Library[1], National Library of New Zealand[2], National Library of Norway[3], Digital Vatican Library[4], Gallica[5]), and exclusively digital libraries have emerged (Digital Public Library of America[6], Europeana[7], HathiTrust[8]).

Positioned at the intersection between digital collections and the researchers who use these collections, libraries are confronted with the problem making the digitized objects accessible. A number of libraries and archives have taken on Machine Learning as an approach to address either lack of metadata or, more generally, the problem of discovery for vast collections in an online search environment (Griffey, 2019; Padilla, 2019; Cordell, 2020), (also see US Library of Congress Newspaper Navigator[9], "Living with Machines"[10], Indiana University Libraries[11]).

While the digital format (text, image, audio, video) is important in libraries, discovery is dependent upon organization by subject area, including field of study. Librarians and specialists in units across the library fulfill the essential tasks of selection, acquisition, and collection development. Their domain expertise is critical to a contextual understanding of library materials which ultimately informs the selection and evaluation of data which is fundamental to Machine Learning processes. As Boyd & Crawford (2012) pointed out, context is essential to deriving meaning from the analysis of data. The full extent of the decision making that goes into data curation seems to be poorly understood as evidenced by a recent paper that suggests re-creating the discipline of Information Science as a subfield of machine learning (Jo &

---

*Both authors contributed equally. [1]Stanford University, California, USA. Correspondence to: Claudia Engel <cengel@stanford.edu>.

*Proceedings of the European Conference on Machine Learning 2020, Teaching Machine Learning Workshop*, Ghent, Belgium, ECML-PKDD 2020. Copyright 2020 by the author(s).

[1]https://wellcomelibrary.org/what-we-do/digitisation/

[2]https://natlib.govt.nz/collections/digital-preservation
[3]https://www.nb.no/en/digitizing-at-the-national-library/
[4]https://digi.vatlib.it/
[5]https://gallica.bnf.fr/
[6]https://dp.la/
[7]https://www.europeana.eu/
[8]https://www.hathitrust.org/
[9]https://news-navigator.labs.loc.gov/
[10]http://livingwithmachines.ac.uk
[11]https://news.iu.edu/stories/2018/11/iub/releases/01-libraries-receives-mellon-grant-to-search-digitized-a.html

Gebru, 2020).

Whether the end goal is injecting Machine Learning into existing processes or developing entirely new processes to take advantage of the affordances of AI, the expert knowledge of information professionals is needed in the design of these efforts. Individuals in roles across libraries, archives, and museums need to understand how AI technology works and the kind of tasks it is well-suited to. And yet retraining all of those professionals as machine learning engineers would not only be costly, but impractical and inefficient. The 2019 Machine Learning + Libraries Summit organized by the US Library of Congress concluded that "there is much more 'human' in machine learning than the name conveys" (Jakeway et al., 2020; Nelson, 2020; Boykis, 2019).

With regard to knowledge building we are thus faced with two issues: (a) many who will not engage directly in computational AI work need a functional understanding of the processes and the work involved and (b) available AI training focuses on the mechanics and theory of the technology itself, catering primarily to an audience of technologists and programmers.

## 2. Pilot experiences

We implemented two training opportunities, geared towards beginners in AI, one at an international conference and another one at our own institution.

### 2.1. Conference workshops

We planned a series of workshops for a conference on the use and implementation of AI in libraries, archives and museums[12] with the aim to provide participants with tools (in the broadest sense) and strategies they would be able to take back into their daily work. As one of the members of the planning committee put it: "We want them to go home with something they can immediately apply." The audience included stakeholders from academia, libraries, industry and society.

The initial challenge was catering to a diverse audience with very different skill sets. For example, in a pre-conference survey of about 100 potential attendees 60% said they would attend a workshop that requires python knowledge vs. 40% that would not. In order to address this, workshops were held on two separate days. One day would lay a foundation for technologies like python, IPython notebooks and TensorFlow, and a second day built on those skills by presenting case studies, often around a particular project and application of AI which was presented by leaders of these projects. We also surveyed attendees regarding topical interests. Those built the structure for the second day, which

ran in parallel tracks, each of which addressed a particular subtopic (text, audio, images, maps, administration). Instructors were recruited from the pool of conference attendees. We deliberately intended to tap into the diverse offerings of their various projects. The design of the individual workshops was left to the instructors.

Asked if workshop participants took something away from the workshops to apply or build on in their work, out of a total of 35 responses all responded with either 'yes' (71.9%) or 'maybe' (28.1%). Respondents appreciated the technical training provided, in particular "a very first experience of elements of coding" or the "concrete technical details about how to build an AI-based application", or "methods and sample code to get started". Participants also found "inspiration for future project ideas".

An important aspect was "meeting others with similar forward thinking ideas and vision", "networking with colleagues and greater awareness of parallel projects at other institutions", a "greater sense of community interest in this new field ..." and "people I can reach out to for help".

Participants appreciated insights into the complexity of the topic:

> "Brainstorming and brain-writing for project planning is much more difficult for new users than I thought previously"

> "I definitely *was* convinced that this was easier than I thought."

> "... the technology and new techniques need unpacking first."

Quite a number of participants wished for more examples to apply AI, including case studies:

> "I really liked the hands-on pieces, but I'd love to see/hear more about what people are doing with the results and how they're being integrated into larger systems or deployed at scale."

> "Will be great to see more case studies in the future as people actually try out this stuff!"

### 2.2. AI study groups

A group of library staff who completed the Elements of AI[13] course all indicated in a subsequent survey that they found the course worthwhile and would recommend it to others but they also expressed the desire for a different kind of instruction. Elements of AI is a free online course, open to anyone. No coding skills or other previous understanding of AI are required for participation. The aim of the project

---

[12]http://fantasticfutures.stanford.edu

[13]https://www.elementsofai.com/

is to teach the basics of artificial intelligence to 1% of EU citizens by 2021.

Library staff found the course material to be too general in its application. At the same time, they found the presentation of historical and philosophical concepts too narrowly confined within the domain of computer science. As information professionals, they recognized roots of many of the core concepts — pattern recognition, grouping, and classification — within their own work. But explanations of these concepts within the course began and ended with research within AI itself. They consistently recommended that examples or experiments of using AI in libraries be included to make it more relevant.

Study groups were offered to give course participants an opportunity to meet regularly and share ideas as they progressed through the material. The weekly reviews and discussions of the material significantly improved participation as well as interest in continued study. As one participant stated:

> "The group discussions were helpful in making the material relevant to both the libraries and our individual interests. Following the material on my own would have been fine, but the virtual face-to-face engagement across different library departments gave me much greater perspective than if I had just read through it on my own."

Though the study groups did not include any formally arranged supplementary readings, the conversation tended to orient the material back to potential applications within the library. By the end of the course, participants in the study groups were eager for both thought exercises and examples of actual applications of AI within the library.

> "..it would be good to focus the group discussions on how each chapter's points or ideas are relevant... For example, we learned about the nearest neighbor concept this week - how could that be used in the library?"

Finally, the participants expressed an interest in understanding the implications of AI, not only within libraries, but in society:

> "I would incorporate a way to demonstrate the usability of AI into the workflows and understand its implications."

> "I also would have liked to learn more about the cost of AI, both socially and financially."

## 3. Takeaways

Based on feedback from our participants in the training offered, the following areas emerge as essential to teaching AI and Machine Learning:

### 3.1. Domain relevance and applicability

There is a strong desire to understand what AI can and cannot do; what the relevance is for one's own field. For example, while the Elements of AI course provided a general overview of AI concepts, participants wanted examples and questions more directly related to their own domain. Participants of the conference workshops were interested to learn "how AI can speed up traditional scientific research" and while they were appreciating the variety of case examples presented in the workshops they were also asking for more.

### 3.2. Broader implications

For librarians who see this technology not as an end in itself but as an instrument to help them fulfill needs of their work, there is a desire to understand AI in a social, political, and technological context. When training experiences were too narrowly focused on AI as technology, participants wanted to understand the broader implications. Participants in a recent AI workshop expressed interest in acquiring "competency to read a study and understand the method" and wanted to be able to evaluate "how trustworthy ... a study [is]". Ethical considerations, central to information management in the library, are important as well. Respondents from our pre-conference survey stated:

> "I hope to bring together the worlds of data and computing and I think discussions about how to do so ethically are really important."

> "[I am interested in] learning how the values that we hold as libraries will shape how AI benefits us, especially if we libraries are active participants in the landscape of AI."

### 3.3. Building intuitions

One of the most instructive exercises in the study groups was a supplemental exercise, inspired by the sheltering-in-place imposed as a protection from the spread of COVID-19, using the "Don't Touch Your Face" online app[14] to demonstrate the gathering of training data and training of a model.

The exercise helped participants leap over both the theoretical concepts and the programming obstacles to learn directly from experience. Watching, in real time, the creation of two classes of training data helped them intuit both how a model is trained and what can go wrong if the training data is poor

---

[14]https://donottouchyourface.com

or the model is inadequate for the task. For example, a participant asked: "Will it know if your hand is in front of your face, but not actually touching your face?" That question presented an excellent teaching moment to address the complexity of potential failure: from the limits of machine perception to a poorly defined training data set.

### 3.4. Collaboration

AI is a multifaceted field where interdisciplinary partnerships are encouraged. And yet collaborative settings and networking are not only important for improved outcomes. The value of collaboration in both learning and problem-solving is too often overlooked. Implicit in the training experiences shared above is a spirit of collaboration, and important facilitator for learning, as evidenced in the success of the study group. As one conference workshop attendee put it:

> "I look forward to getting inspiration from others and would also hope to start collaboration with some libraries or other players."

## 4. Discussion: What to teach and how

### 4.1. What

Too often AI curricula are determined based on the student's level of programming experience. Our experience indicates that training focused around the uses, benefits, and potential harms of the technology are essential regardless of the technical knowledge.

We also propose locating training within the context of a domain. As noted in the Machine Learning + Libraries Summit report "when discussing implementation steps, domain-specific knowledge and expertise of staff was mentioned more often than ML and AI" (Jakeway et al., 2020).

Based on these considerations we developed the following learning goals:

- Outline the current landscape of relevant, state of the art AI projects

- Articulate the problem to be addressed with AI and the expectations from the process

- Describe the steps needed to prepare training data, to train, evaluate and deploy a model, to use a pre-trained model.

- Assess ML methods, approaches, tools with respect to relevance for particular projects and evaluate outcomes: What are tasks and processes that can be modified, improved or scaled?

- Address the implications of data-driven models through critical consideration of the source of the data, the context, the bias, the relevance to the question to be answered

- Evaluate data sets with regard to their value for model training / for an ML approach, including concerns of bias

- Analyze the shape of the data to learn where there may be gaps, anomalies, or recognizable patterns

- Articulate how bias can affect applications of AI and state ethical implications of AI in your organization and beyond

### 4.2. How

We have identified several examples that we propose to build upon to establish training methods. Both of the approaches described below help to fill the void of practical experience for non-programmers who nonetheless want to dig in and learn.

Recently, we taught an introduction to AI in the form of a guided question and answer session. Concepts were described in a group conversation and accompanied by the corresponding python code. The goal was to demystify the workings for non-programmers by revealing the few lines of code that make it possible to implement and execute the concept. Similar micro-training examples in the form of focused simulations have been used to help communicate the relationship between code and results both for communicating the usefulness of commercial products[15] and communicating about new machine learning techniques[16].

Another model we have begun to explore more deeply is to have training and learning embedded in actual projects by making use of notebooks for exploratory programming, visualization and documentation. Fast.ai, for example, bases their online training entirely on notebooks[17], which are versatile training tools. We have also seen evidence that sharing notebooks can seed collaboration and contribute to much needed discussions around the adoption of AI in academic research. Much can be learned from case examples and from studying ongoing projects.

Overall, we see Machine Learning as inspiring new methods and accelerating essential work for which implementation requires thoughtful decisions about data, models, and assessment of outcomes, which in turn requires a range of expertise and competencies. Along with the necessary skills to do the computational work goes the expertise to understand the implications of developing and training models,

---

[15] http://streamlit.io/
[16] https://distill.pub/
[17] https://www.fast.ai/2019/12/02/nbdev/

and their interpretation and evaluation, a process within which librarians, technologists, and researchers all have important roles.

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
