# OpenReview forum: "”AI is not Just a Technology”"
_ECMLPKDD.org/2020/Workshop/TeachML — ECML PKDD 2020 TeachML_

### Official Review · AnonReviewer2 · 2020-07-15
**An inspiring paper about teaching AI in a specific context**

**Rating:** 9
**Confidence:** 4

**Review:**

The paper discusses how to teach AI in the context of libraries. It discusses two trainings that were implemented and evaluate what learners thought about the courses and what the take aways are from the learner's feedback.
I found the paper very inspiring and fitting for this workshop as it not only describes who the authors teach AI and ML, but also reflects on practices and how it can be done better such that the learners get the most out of the learning experience.

## Pros:
A well written paper with interesting examples. Although it is focused on teaching library employees, I believe the learnings are applicable to a broader audience. I even leared a lot from the references (Elements of AI). In particular the takeaways were interesting to me and probably will be to others. I will aim to focus my teaching "around the uses, benefits, an potential harms of the technology", as the authors suggest.

## Cons:
Only parts of the teaching materials are made available for the reader.
I would like to see the entire material and know under which license it could be reused.

### Minor comments
- I wonder if the title should reflect that the paper is centered on teaching people working in libraries (on the other hand this might stop people from reading it, so not really sure)
- Section 1, line 033: "(Boyd & Crawford, 2012)" -> "Boyd & Crawford, 2012"
- Section 4.2, line 186, typo: "conversation group conversation"

---

### Official Review · AnonReviewer1 · 2020-07-26
**Cross-pollinating AI education with domain-specific questions for academic libraries**

**Rating:** 7
**Confidence:** 5

**Review:**

The paper summaries the authors' findings on introducing staff of academic libraries to AI. This problem is, in many ways, domain specific as many obvious applications of AI can be identified in advance (as opposed to designing a curriculum for CS students which cannot make such assumptions). Having said that, knowledge discovery strikes at the very heart of what AI should be used to do and I felt the paper presented some very interesting ideas.

The authors present findings from two training experiences aimed at beginners in two different venues.

The collaborative experience in the conference workshop setting is an interesting idea and one that deserves follow-up. The audience, despite the challenges with catering to a diverse set of people, seems to have responded positively to the training offered. This seems to be evidenced by an overwhelming positive response. However, the number of participants (and their backgrounds) would help place this better in context.

The experience with the local AI groups likewise seems to have built positively on top of a general introduction to AI. This, in itself, is hardly surprising; it would be more interesting to see the medium- and long-term effects of these trainings. For instance, interviews with course participants some months down the line to see if they were able to incorporate some of the introduced topics into their daily workflows or whether they were able to follow through on the collaborative projects mentioned by them.

I would also encourage the authors to consider metrics that can be used to quantify the results of their experiments in a data-driven way that goes beyond qualitative results.

---

### Official Review · AnonReviewer3 · 2020-07-28
**Teaching Machine Learning for subject matter experts**

**Rating:** 6
**Confidence:** 5

**Review:**

This paper reports experiences with teaching machine learning topics to library staff. The authors describe very well the motivation to apply machine learning technology to the collected knowledge of the libraries.
The idea to spark interest in non-technical staff by teaching them programming and machine learning topics is appealing. However, the question remains if this approach is beneficial in the long term as there is a steep learning curve for beginners to develop correct and working AI projects.
The four main takeaways as they are described in the paper, all seem relevant and reasonable. These aspects are useful for AI experts teaching this topic and could improve the experience for all kinds of learners.
The collected qualitative feedback is interesting and helpful to improve the content of the training. Yet quantitative data would also be interesting to answer questions. For example, if the training attendees could apply the new knowledge in their daily work.

As a person that also teaches ML, I would have liked to read more about the structure and the content of the workshops. The described ideas are interesting but very general.


Minor issues:
Typo Line 185, Row 2 "a conversation group conversation"

---

### Decision · Program_Chairs · 2020-07-31

**Decision:**

Accept

**Comment:**

The reviewers agree that this paper will be accepted. Thank you for your contributions.

Please register with the conference as soon as possible! See this page for details:
https://ecmlpkdd2020.net/attending/registration/.
Which asks that at least one author per paper registers until July 31, 2020.
We apologize for the very short notice.